# Clinical Approaches for the Management of Skin Cancer: A Review of Current Progress in Diagnosis, Treatment, and Prognosis for Patients with Melanoma

**DOI:** 10.3390/cancers17040707

**Published:** 2025-02-19

**Authors:** Colton Connor, Quinton L. Carr, Alisa Sweazy, Kelly McMasters, Hongying Hao

**Affiliations:** 1School of Medicine, University of Louisville, Louisville, KY 40202, USA; colton.connor@louisville.edu (C.C.); qlcarr01@louisville.edu (Q.L.C.); 2The Hiram C. Polk, Jr., MD Department of Surgery, School of Medicine, University of Louisville, Louisville, KY 40202, USA; alisa.sweazy@louisville.edu (A.S.); mcmasters@louisville.edu (K.M.)

**Keywords:** melanoma, diagnosis, treatment, prognosis

## Abstract

Melanoma remains a significant cause of skin-cancer-related mortalities globally and incidence rates are rising. Advances in diagnostic technologies have led to less invasive methods for biopsies and more personalized risk assessments based on tumor molecular profiles. Treatment modalities have progressed to include targeted therapy and immunotherapy, but challenges with resistance and toxicities remain. The intent of this review is to provide an overview of the current clinical state of the disease, specifically in regard to the recent progress made in the areas of diagnosis, treatment, and prognosis. Existing challenges and the direction of future research will be outlined.

## 1. Introduction

Melanoma represents a significant public health challenge due to its increasing incidence and potential for metastasis. While only accounting for a small percentage of all skin cancers, melanoma remains the number one cause of skin-cancer related mortalities worldwide [1]. Skin cancer can be divided into two categories: non-melanoma skin cancer and melanoma [2]. The primary non-melanoma skin cancers include squamous cell carcinoma (SCC) and basal cell carcinoma (BCC) [3]. This review article will specifically address melanoma and the latest advancements made in its diagnosis, treatment, and prognosis. At current incidence rates, it is estimated that the number of new cases of melanoma will increase by 50% by 2040, leading to a 68% increase in mortalities based on data from 2020. Australia currently leads with the highest incidence rate of melanoma worldwide, with Europe and North America representing the second and third highest incidence rates, respectively [4]. While this can be attributed to the high-risk populations in these regions, another contributing factor to the increase in melanoma incidence may be the recent awareness of the cancer and its onset overdiagnosis through the use of proxy screens [5,6]. Regardless, one of the most important causes of melanoma is ultraviolet (UV) radiation [7]. UV radiation can lead to degenerative aging, inflammation, carcinogenic mutations, and suppression of various aspects of the immune system [8]. It has been shown that lighter-skinned populations, those of European origin, are more susceptible to UV radiation and thus more likely to develop skin cancer due mainly to decreased levels of melanin and a lower amount of the photoprotective eumelanin in proportion to pheomelanin [9,10]. Importantly, darker-skinned populations are still at significant risk of the damaging effects of UV radiation, such as photo-dermatoses and photo-exacerbated pigmentary disorders [11]. Other risk factors for the development of melanoma include genetic predisposition, advanced age, many nevi, and immunosuppression [9,12,13,14].

Despite the increasing incidence of melanoma, mortality rates have declined slightly year over year since 2015 [1]. A recent study reports a 25.2% decline in mortality from melanoma from the 1970s to the 2010s, with increased risk for mortality being associated with Breslow thickness, nodular subtype, age at diagnosis, and anatomic location [15]. Methods for early detection and accurate staging have been enhanced by new diagnostic strategies and contributed to the decrease in mortality rates of melanoma. Populations are also more aware of the risks of acquiring melanoma and are more active in preventative measures, such as the use of sunscreen to protect against UV light [16].

Likewise, treatment modalities have expanded beyond traditional surgical excision to include targeted therapy and immunotherapy. These treatments have transformed the therapeutic landscape and improved patient outcomes in recent years. Advances in understanding the molecular characteristics of melanoma have led to the identification of new therapeutic targets [17,18]. As technologies like genetic sequencing and bioinformatics continue to evolve, immunotherapy and targeted therapy are expected to become even more effective, enabling personalized treatment strategies tailored to individual patients [1,19].

Prognostic assessment has benefited from the development of novel biomarkers and genetic profiling. The prognosis of cutaneous melanoma depends mainly upon tumor thickness, ulceration, and lymph node status, but is also affected by age, sex, and anatomic location of the primary tumor. Progress has been made on more precise and individualized risk stratification to guide personalized medicine. Recent improvements in early diagnosis, customized prognosis, and targeted treatment have led to increasing rates of survival [20,21]. This review will address the recent progress made in the areas of diagnosis, treatment, and prognosis for patients with melanoma.

## 2. Diagnosis

Melanoma is an aggressive form of skin cancer, with malignant melanoma being the leading cause of mortality among skin cancers [1]. A broader concept of melanoma diagnosis includes initial diagnosis of melanoma and diagnosis of melanoma at recurrence. After a biopsy establishes the initial diagnosis of melanoma, several clinicopathological factors, such as primary tumor site (extremities, trunk, head and neck), tumor thickness, primary tumor ulceration, lymph node status, and signs of distant metastasis are evaluated to determine the stage of melanoma. The currently used staging system is the 8th edition of the American Joint Committee on Cancer (AJCC) melanoma staging system [22,23,24]. Accurate staging at melanoma initial diagnosis is important to guide the individualized disease management.

Even though patient outcomes have been improved drastically with the progress on targeted therapy and immunotherapy, many melanoma patients still develop recurrent disease. Imaging modalities and molecular diagnostics, such as circulating tumor DNA (ctDNA), have been investigated for diagnosis of recurrent melanoma [25,26]. Advanced imaging techniques and molecular characterizations bring new hope for more accurate assessment. However, despite interest in improved diagnostics, histological and immunohistological techniques remain the gold standard for the diagnosis, grading, and staging of melanoma [27].

### 2.1. Advanced Imaging Diagnostics

#### 2.1.1. Digital Dermoscopy

Enhanced imaging modalities such as digital dermoscopy (DD), also known as epiluminescence microscopy (ELM) or surface microscopy, provides a non-invasive method by which to observe and detect morphologic changes in cutaneous tissue. For instance, lentigo maligna, the most common histologic melanoma subtype, can be differentiated based on its annular granular pattern, angulated lines, and confluence of perifollicular pigment on dermoscopy, highlighting the utility of this diagnostic approach [28,29]. DD uses epiluminescence light microscopy to magnify the skin lesions and observe the dermo-epidermal junctions [28,29]. DD has been shown to reduce the number of unnecessary biopsies [30,31], with one study reporting 42% fewer excisions with dermoscope use [32]. Sequential DD has been indicated for high-risk patients and can be performed in short-term intervals of three months for fast-growing lesions and long-term intervals of six to twelve months for multiple, non-suspicious lesions, thus suggesting that this method of surveillance enhances both sensitivity and specificity in detecting melanoma in high-risk individuals [33]. However, for low-risk patients, sequential surveillance is not recommended [34]. It has been shown that for the low-risk population, a single dermatoscopic examination is sufficient to identify melanoma, with no significant difference between tumors diagnosed by periodic use of a handheld dermoscope compared to sequential DD [35,36]. When combined with computer vision systems, DD can increase diagnostic accuracy for melanoma [36]. Recent studies have also suggested that dermoscopy can aid in identifying small-diameter melanomas, specifically flat, non-facial melanocytic lesions measuring ≤5 mm [37]. A predictive model using an atypical pigment network, blue-white veil, pseudopods, peripheral radial streaks, and multicolored characterization as dermoscopic predictors was able to identify melanoma with a longitudinal diameter of <5 mm with 65% sensitivity and 86.4% specificity [37]. However, dermoscopic images may have some deviations and artifacts. For this reason, DD should be combined with other diagnostic measures such as histology or immunohistochemistry to increase diagnostic accuracy [29].

#### 2.1.2. Line-Field Confocal Optical Coherence Tomography

Optical coherence tomography (OCT) and reflectance confocal microscopy (RCM) are non-invasive techniques used for the diagnosis of skin cancers such as melanoma. The lower resolution of OCT results in less cellular resolution compared to RCM but has greater depth penetration, allowing for the creation of vertical images at the mid-to-deep dermal level that can be used to measure tissue structural patterns and tumor thickness [38,39]. This aids in the early detection of melanoma. RCM produces images in the horizontal plane and is well suited for determining tissue morphology, which is useful in the detection of atypical cells and analyzing pigmented lesions and aids in the determining benign from malignant features [40,41]. RCM can help provide a clearer definition of lesion boundaries, thus enabling a more precise and effective diagnostic approach and is especially useful for identifying the histological basis of dermoscopic features in facial pigmented lesions [42]. While these diagnostic approaches are useful for specific cases, OCT and RCM are not routinely used for melanoma diagnosis and are more often employed as adjunctive techniques that serve as supplementary tools rather than replacements for biopsy and histology. Line-field confocal optical coherence tomography (LC-OCT) is a novel technology introduced in 2018 that combines the benefits of both OCT depth acquisition and RCM isotropic resolution, providing high-resolution, cross-sectional images of cutaneous structures that are then converted to three-dimensional cubes/slices [43,44]. LC-OCT has been shown to have increased diagnostic accuracy compared to dermoscopy, although this stems from its ability to rule out malignance rather than make an exact diagnosis [45]. In a study by Schuh et al., the LC-OCT device showed improved distinction between nevi and melanomas when analyzing melanocytic lesions, leading to better diagnoses and reducing the occurrence of unnecessary surgical biopsies [46]. Although LC-OCT presents a significant benefit to the techniques used for the diagnosis of melanoma and other skin cancers, further research is required to confirm the existing data for this novel technology [47].

#### 2.1.3. Diagnostic Imaging Techniques Using Radiolabeled Probes

When addressing the metastatic concern for melanoma, many diagnostic modalities may be employed based on the patient’s stage and clinical context. Presently, lymphoscintigraphy followed by sentinel lymph node biopsy (SLNB) is commonly performed to identify early melanoma metastasis to local lymph nodes. For distant metastases, computed tomography (CT), positron emission tomography (PET), and magnetic resonance imagining (MRI) may be used [48]. Recommendations for the appropriate diagnostic and follow-up use of these imaging tools may be based on cancer stage. Recently, the American Joint Committee on Cancer (AJCC) divided invasive melanoma into a “low-risk” (stage IA–IIA) group and “high-risk” (stage IIB–IV) group [22,23,24]. A collaboration of European experts in 2022 recommended whole-body cross-sectional CT or PET-CT scans combined with MRI for patients with stage IIC metastatic melanoma [48]. Newer imaging agents, such as melanin-targeted radiotracers or nanoparticles, are being researched and developed to improve the sensitivity and specificity of melanoma imaging with PET, particularly in detecting metastases and assessing the extent of disease. These radiolabeled probes were developed to target melanoma-specific markers like the melanocortin-1 receptor (MC1R) and melanin itself and have demonstrated specific uptake in melanoma tumors, which helps in tumor visualization while minimizing off-target effects in normal tissues [49]. Additionally, newer theranostic approaches combine imaging and treatment by using radioisotope-labeled antibodies that target melanin, such as 203Pb and 212Pb-labeled melanin-specific antibodies. These antibodies have shown promise in both imaging melanoma metastasis and treating tumors via radiotherapy, which reduces tumor growth and increases survival rates in preclinical studies [50]. These innovations represent significant progress in the non-invasive detection of melanoma, providing more precise diagnostic options compared to traditional methods.

#### 2.1.4. Dermoscopy Image Analysis Using Machine Learning (ML) and Deep Learning (DL)

Machine learning and deep learning used in the analysis of dermoscopic images have become pivotal methods to assist clinicians in the diagnosis and prognosis of melanoma [51,52]. These advanced artificial intelligence (AI) technologies are able to identify important characteristics of skin lesions, including border and color irregularities, asymmetry, and texture patterns to identify these lesions as benign or cancerous [53]. Specifically, supervised ML models are trained to recognize cancerous lesions based on dermoscopic datasets and can significantly improve the diagnosis and prognosis of melanoma by decreasing detection time and increasing diagnostic accuracy. Such supervised learning models might include Support Vector Machines (SVMs), Logistic Regression (LR), and k-Nearest Neighbors (KNN), to name a few [53]. Deep learning models such as DenseNet and deep convolutional neural networks (DCNNs) have shown superior performance compared to traditional diagnostic methods, achieving over 95% accuracy on complex datasets [53]. Ultimately, the use of ML and DL can provide valuable insights for diagnosis and clinical decision-making for patients with melanoma, thus highlighting the necessity of integrating advanced AI technologies into the diagnostic process to improve patient outcomes. However, the use of AI for the diagnosis of melanoma is not void of challenges, including the need for more diverse datasets, better methods for model interpretation, and more efficient integration into clinical workflows. Future research should place emphasis on overcoming these obstacles.

### 2.2. Molecular Diagnostics

#### 2.2.1. Circulating Tumor DNA (ctDNA)

Another method for diagnosing melanoma utilizes ctDNA as a form of non-invasive liquid biopsy. Though ctDNA is not specific to circulating cancer cells (CTCs), CTCs are known to produce ctDNA. CTCs can be identified by markers like melanoma-associated antigen 3 (MAGE-A3), melanoma cell adhesion molecule (MCAM), melan-A, and chondroitin sulfate proteoglycan (MCSP), which can provide insights into tumor progression and metastatic potential. Moreover, the presence of PD-L1 expression on CTCs can indicate how a patient might respond to immunotherapy [25]. For melanoma patients on immune checkpoint inhibitors, assessment of plasma ctDNA level based on patients’ specific mutations has been shown to be a reliable method for assessing treatment response and detecting early signs of relapse [26]. The findings in a study by Váraljai et al. support the analysis of ctDNA as a sensitive tool for the treatment response and future progression of melanoma, specifically uncovering a significant correlation with tumor stage [54]. In this study, plasma-derived levels of ctDNA from assays for *BRAF* and *NRAS* driver mutations as well as *TERT* promotor mutations were analyzed in patients with advanced melanoma. It was also shown that ctDNA proved to be a better indicator of tumor stage than the routinely used tumor markers lactate dehydrogenase (LDH) and serum S100 levels [54]. While providing a useful approach for the early diagnosis of melanoma [25], the differentiating characteristic of ctDNA analysis lies in its ability to monitor the therapeutic response to treatment [54,55]. The use of ctDNA remains investigational.

#### 2.2.2. Gene Expression Profiling (GEP)

Today, methods that complement histopathological biopsy evaluations include immunohistochemistry (IHC), fluorescence in situ hybridization (FISH), single nucleotide polymorphism (SNP) array, comparative genomic hybridization (CGH), gene expression profiling (GEP), and next-generation sequencing (NGS) [56]. As ancillary tests, these diagnostic tools are not recommended to replace traditional clinical and pathological evaluations but rather to supplement the patient’s clinicopathologic findings to build a more personalized diagnosis and treatment plan [57]. Like many other types of cancer, melanoma has multiple genetic variations. The Bastian group recently analyzed the genetic features of melanoma [57,58,59]. According to genetic changes and various evolutionary pathways, Bastian et al. showed that melanoma can be classified into low- or high-cumulative-sun-damage melanoma, desmoplastic melanoma, spitz melanoma, acral melanoma, mucosal melanoma, uveal melanoma, and melanoma that arises in congenital nevi or in blue nevi [59]. GEP has particularly received traction in the last decade, helping to predict factors of prognosis, such as the risk of metastasis or recurrence, by focusing on the molecular makeup of the tumor and providing more personalized risk assessments [57,58,59,60]. GEP ancillary molecular profiling tests have been demonstrated to be effective at helping clinicians stratify ambiguous lesions or determine melanoma from a benign nevus [61]. GEPs can identify various numbers of gene signatures, for instance, the 2-GEP (PLA, DermTech, La Jolla, CA, USA) which uses a non-invasive adhesive patch as the sample collection platform, or the 31-GEP test, known as DecisionDx-Melanoma (Castle BioSciences, Inc., Friendswood, TX, USA) [61,62]. However, while these adjunct tests may facilitate diagnosis, there remains controversy at the level of the National Comprehensive Cancer Network as to the role of GEP, particularly in the prognostication of melanoma [61]. Future studies are needed to understand whether GEP tests can reliably predict melanoma outcomes. Currently, GEP tests to determine prognosis and the need for SLNB remain investigational and should not be used for treatment decisions outside of a clinical trial.

## 3. Treatment

Unlike many other solid tumors, the mainstay of systemic therapy for melanoma includes immunotherapy and targeted therapy, with very limited indications for cytotoxic chemotherapy and radiation therapy.

### 3.1. BRAF Targeted Therapy

*BRAF* mutations are present in roughly one-half of advanced melanomas and can contribute to tumorigenesis via the constitutive activation of the MAPK signaling pathway [63,64]. There has been much interest in therapies targeting BRAF mutation and several have been successfully developed [63,64]. The first of these was vemurafenib, a BRAF kinase inhibitor which targets the most common of the BRAF mutations (BRAF^V600E^) [65,66]. Vemurafenib was shown to lead to improved survival in patients with the BRAF^V600E^ mutation when compared to treatment with dacarbazine, the only FDA-approved chemotherapeutic agent for the treatment of metastatic melanoma at the time of the trial [67]. Shortly after the success of vemurafenib, a second BRAF inhibitor, dabrafenib, was shown to be similarly effective [68].

Though *BRAF* mutations are often present in melanoma, there are many other mutations that can lead to aberrant MAPK/ERK signaling and drive oncogenicity [69]. Thus, inhibitors targeting elements further downstream in the signaling pathway have utility as they can block the activity of a variety of upstream mutations. Trametinib, a selective MEK1/2 (mitogen-activated protein kinase) inhibitor, capitalized on this in 2012 when improved survival over dacarbazine was observed [70]. Though BRAF and MEK inhibitors have been shown to increase survival when administered as monotherapy, treatment resistance is common and typically occurs within 7 months [71]. However, BRAF and MEK inhibitor combination therapy can improve overall survival, reduce the risk of disease progression, improve response rates, and delay the development of treatment resistance [71,72,73,74,75].

Adjuvant BRAF/MEK inhibitor therapy can be considered for melanoma patients with a mutation in *BRAF* [76]. The FDA has approved the adjuvant therapy of dabrafenib plus trametinib for stage III melanoma patients with *BRAF* mutation [76,77]. A 10-year follow-up study showed that adjuvant therapy with dabrafenib plus trametinib was associated with better relapse-free survival and distant metastasis-free survival than placebo among patients with resected stage III melanoma [77]. Another phase III trial of combination of BRAF/MEK inhibitor therapy for metastatic melanoma with the *BRAF* mutation also confirmed the long-term benefit of encorafenib plus binimetinib in unresectable or metastatic melanoma patients with a mutation of *BRAF* [78]. With the development of newer, more effective BRAF and MEK inhibitors, there are increasing options for combination therapy.

### 3.2. Immunotherapy Using Immune Checkpoint Inhibitors

PD-1 (programmed cell death protein-1) is an inhibitory receptor expressed on activated T-cells. When bound to programmed death-ligand 1 (PD-L1) or programmed death-ligand 2, PD-1 acts to inhibit T lymphocyte proliferation, dampen effector functions, and reduce survival via signaling through the SHP1/2 pathway [77,78]. PD-L1 is upregulated in response to proinflammatory cytokines in many healthy tissues. PD-L1 is also upregulated in melanoma tumor cells and is thought to act as a mechanism of immune evasion [76,79]. Nivolumab and pembrolizumab are anti-PD-1 monoclonal antibodies. These antibodies work by preventing the PD-1 interaction with its ligand, thereby maintaining T lymphocyte function against melanoma tumor cells [80]. In December of 2021, the FDA approved the use of pembrolizumab for adjuvant treatment of stage IIA or IIB melanoma following surgical resection [81]. In recent phase 3 trials, Nivolumab continued to display efficacy and was approved by the FDA in October of 2023 for the adjuvant treatment of stage IIB/C melanoma following full resection in patients 12 years and older [82,83]. Other clinical trials have shown that adjuvant immune therapy with nivolumab or pembrolizumab has significant benefit for recurrence-free survival and distant metastasis-free survival in stage III melanoma [84,85,86,87,88]. Thus, the current guidelines have suggested that adjuvant immunotherapy should be considered in stage IIIB through stage IIID melanoma, while using adjuvant immunotherapy in stage IIIA remains controversial [89].

In addition to immune checkpoint inhibitors targeting PD-1, anti-PD-L1 monoclonal antibodies can prevent the inhibitory effects of PD-1/ligand interaction in a similar fashion. One anti-PD-L1 monoclonal antibody, atezolizumab, has recently been proposed as a possible addition to vemurafenib–cobimetinib combination therapy, although hesitation from the ASCO concerning toxicity of the drug still remains [90,91].

Ipilimumab is a cytotoxic T lymphocyte-associated antigen 4 (CTLA4) inhibitor [92]. CTLA4 inhibitors work by a mechanism similar to those of anti-PD-1/anti-PD-L1 monoclonal antibodies. Like the PD-1 receptor, CTLA4 is an immune inhibitory receptor that, when bound by its ligand, B7, inactivates T lymphocytes and allows tumor cells to escape the immune response [84]. Though survival benefits have been seen with ipilimumab, the response rate is low, and high toxicity is associated with treatment [85,86,92]. Though therapy is more effective when ipilimumab is combined with nivolumab, toxicity remains a significant issue [87,88,89,93]. Combining standard doses of agents which block the PD-1/PD-L1 axis with low doses of ipilimumab may potentially reduce toxicity while maintaining increased effectiveness over monotherapy with either drug class [94,95,96,97,98]. Furthermore, combination therapy with anti-PD-1 antibodies and CTLA4 inhibitors has demonstrated superior efficacy in the treatment of melanomas in comparison to dual BRAF and MEK inhibition [99]. In recent phase 3 trials, neoadjuvant use of nivolumab plus ipilimumab showed superior results to therapeutic lymph node dissection followed by adjuvant immunotherapy—the current standard of care for resectable, macroscopic stage III melanoma [100]. These findings highlight a major shift in the way melanoma may be treated going forward.

Expert dermato-oncologists from Europe recently released a guideline for melanoma treatment, stating that adjuvant therapies can be applied to completely resected stage IIB through stage IV melanoma. PD-1 inhibitors are approved in stage II patients. Anti-PD-1 therapy or dabrafenib plus trametinib can be considered in stage III patients with *BRAF* mutations. Nivolumab, as well as ipilimumab and nivolumab, can be offered in selected, high-risk, stage IV patients. Neoadjuvant therapy with pembrolizumab followed by complete surgical resection and adjuvant pembrolizumab is recommended in patients with clinically detected macroscopic, resectable melanoma. In patients with irresectable stage III/IV melanoma, first-line systemic treatment using PD-1 antibodies alone or in combination with CTLA-4 or LAG-3 antibodies should be considered [101].

Relatlimab is a lymphocyte activation gene-3 (LAG-3) inhibitor that has been shown to be effective for the treatment of melanoma when used in combination with the PD-1 inhibitor, nivolumab. It has received FDA approval for unresectable or metastatic melanoma. Neoadjuvant relatlimab/nivolumab has shown promising results as well, with an acceptable toxicity profile [102,103].

### 3.3. Tumor-Infiltrating Lymphocyte Therapy

One promising area of research is the use of tumor-infiltrating lymphocyte therapy in melanoma. The presence of tumor-infiltrating lymphocytes (TILs) are a positive prognostic feature in melanomas, as they are thought to represent a natural antitumor response [104]. Therapy with tumor-infiltrating lymphocytes capitalizes on this phenomenon via the expansion of autologous melanoma TILs for reinfusion. Prior to reinfusion, patients are subjected to lymphodepleting chemotherapy, which increases levels of circulating cytokines responsible for promoting antitumor activity of TILs [104,105]. Following infusion, patients are treated with high-dose IL-2, which further boosts the efficacy of the TILs [106]. Benefits of TIL therapy include the ability to selectively amplify lymphocytes with the highest affinity for patient-specific tumor antigens. Furthermore, the autologous nature of TIL therapy helps to avoid issues with rejection [107]. Positive results have been observed with TIL therapy, warranting further investigation [108]. Recently, the U.S. Food and Drug Administration approved the TIL therapy lifileucel for patients with unresectable or metastatic melanoma previously treated with PD-1 inhibitors and, if *BRAF* mutation is present, BRAF/MEK inhibitors. It is the first cellular therapy to be approved for a solid tumor [109,110,111].

### 3.4. Vaccinations

There are several types of melanoma vaccines currently under development. Though melanoma vaccines are still in the early stages of development, the availability of tools such as next-generation sequencing and in silico epitope prediction algorithms allow for the efficient identification of tumor-specific neoantigens, which may ultimately lead towards personalized vaccine therapy [112]. Personal mRNA neoantigen vaccines against melanoma demonstrated promising results when combined with pembrolizumab in recent trials [113,114]. One challenge related to treatment with personal neoantigen vaccines includes delayed treatment as there is an unavoidable waiting period between sample collection and vaccine delivery, though efforts to reduce this waiting period are underway [115].

### 3.5. Gene Therapy

Cells become cancerous when they acquire mutations in signaling pathways that promote unchecked growth. These mutations can reduce the cell’s ability to enact antiviral response mechanisms. In recent years, viruses have been developed to selectively infiltrate cancer cells and exploit this weakness [116,117]. The only virus approved for the treatment of metastatic melanoma is talimogene laherparepvec (T-VEC), a herpes simplex virus vector gene therapy. T-VEC causes tumor lysis via oncolysis and subsequent expression of granulocyte–macrophage colony-stimulating factor (GM-CSF), which helps to recruit and activate antigen-presenting cells and consequently induce a T-cell response [117,118]. T-VEC is a promising option for treatment, as 47% of injected lesions underwent total resolution during the phase 3 OPTiM trial [118]. However, a large, randomized trial found that T-VEC + pembrolizumab was not superior to pembrolizumab alone for patients with stage III and IV melanoma [119]. In addition to T-VEC, several other oncolytic viruses are under development for the treatment of melanomas [120,121,122,123].

### 3.6. Toll-like Receptor Agonists

Toll-like receptors (TLRs) play an important role in activating both the innate and adaptive immune response [124]. TLR agonists have been developed in hopes of improving the body’s immune response towards cancer. When used in combination with other therapies such as BRAF inhibitors, small molecule inhibitors, IL-2, and vaccines, TLR agonists have displayed synergistic effects in the treatment of melanomas [125,126,127,128,129].

### 3.7. Electrochemotherapy

Electrochemotherapy (ECT) is a treatment option which relies on electroporation to increase the uptake of chemotherapeutic drugs, most commonly bleomycin in the case of melanoma [130,131]. ETC has demonstrated favorable oncological outcomes, high response rates, and an ability to aid in local tumor control despite maintaining low toxicity. Additionally, ETC is modifiable in that it can be administered locally or locoregionally depending on whether chemotherapy is injected intratumorally or systemically [130]. Additional survival benefit has been observed when ETC is combined with systemic therapies [132,133].

### 3.8. Radiotherapy

Radiotherapy is a common choice for the treatment of a variety of cancers, although some discussion regarding its efficacy for the treatment of melanoma remains [134,135]. Following the development of immunotherapies, though, promising results have been observed when these treatments are co-administered [136,137]. Combination therapies also appear to increase the likelihood of the abscopal effect, a rare phenomenon that involves the shrinkage of metastatic lesions following the use of radiation at a distant site [138]. With the advent of new therapeutic strategies, radiotherapy may gain increasing utility in the treatment of melanomas. However, this approach remains investigational. Currently, the main utility of radiation therapy for patients with melanomas remains in the treatment of unresectable brain metastasis. Recent findings have introduced the idea of an Anti-Warburg Effect (AWE) in circulating tumor cells, causing them to shift from a reliance on glycolysis to an increased reliance on oxidative phosphorylation. Though this is a recent discovery, clinical implications likely exist, as the AWE is strongly correlated with therapeutic response in melanoma patients [139]. Though the utility of radiotherapy with regard to the AWE has not been fully established, this discovery may lead to the increased use of radiotherapy with improved understanding.

Several confounding factors exist in the treatment of melanoma (Figure 1). Lifestyle, genetic heterogeneity of tumors, immune suppression status in the tumor environment, and changing metabolic settings may affect the treatment efficacy. Controversy remains regarding the links between obesity and the efficacy of targeted therapy and immunotherapy [140]. Diet can reshape the gut microbiome, affect the immune and the metabolic landscape of the tumor microenvironment, and therefore modulate the efficacy of immunotherapy [141,142,143]. Significant outcomes are seen in patients with “hot” tumors that have an active immune signature. However, in patients with “cold” tumors, the cancer demonstrates an immune-desert tumor profile, and treatments often end in non-responsiveness [144]. Determining how these personalized confounding factors should be considered to improve therapeutic efficacy remains a significant challenge in this field.

## 4. Prognosis

The eighth edition of the AJCC melanoma staging system that was implemented in 2018 provides a useful approach to melanoma staging at initial diagnosis. This commonly accepted classification system allows for more precise risk stratification for melanoma prognosis [22,23,24]. As reflected in the AJCC staging system, the prognosis of melanoma depends on a number of histologic and clinicopathologic features. Breslow tumor thickness is typically considered one of the most significant prognostic indictors, although many clinical elements at diagnosis influence potential outcomes, including ulceration, lymph node involvement, metastasis, mitotic rate, genetic mutations, age and gender, tumor location, and immune response. The prognosis of metastatic melanoma has improved since the approval of ipilimumab in 2011 and pembrolizumab in 2014 [145,146,147]. Additionally, several recent clinical trials support the use of immunotherapies and combination regimens to increase overall survival rates for patients with unresectable advanced melanoma [88,148]. Neoadjuvant therapy for stage III melanoma patients has been suggested by the results of some clinical trials. However, there is still a need to understand the relationship between pathological response, recurrence-free survival (RFS), and overall survival (OS) in patients with this disease [149]. Newer research is focusing on biomarkers and other genetic and immune-related factors that can provide more accurate predictions of prognosis.

Prognostic calculators have been developed to stratify risk in melanoma patients. A couple of well-known prognostic factors, such as sentinel node status, Breslow thickness, presence of ulceration, age at SLNB, primary tumor location, and maximum diameter of the largest sentinel node metastasis, were incorporated in the model [150,151,152]. These models can accurately predict patient-specific risk probabilities for 5-year recurrence-free and melanoma-specific survival [152]. These tools offer clinicians to make better decisions to select tailored adjuvant treatments.

Predictive biomarkers for melanoma immunotherapy generally can be divided into tumor-related factors, host immune responses, and tumor microenvironment characteristics. Recent research on PD-L1 expression and its ability to reflect the immune status of the body has indicated that the biomarker may have poor predictive power for the prognosis of immunotherapy [88,153,154]. Tumor mutational burden (TMB) as a biomarker has been proven to be a better predictor of prognosis than PD-L1 while also correlating well to the efficacy of immune checkpoint inhibitors [155,156]. Multi-biomarker models that work to integrate various molecular factors of the cancer are expected to emerge as a possible solution to better predict melanoma prognosis [157]. Such biomarkers might include PD-L1, TMB, LDH, tertiary lymphoid structures (TLSs), tumor-associated macrophages (TAMs), tumor-infiltrating lymphocytes (TILs), and extracellular vesicles (EVs). Immunohistochemical markers such as S100, SOX-10, and MITF may present a new direction of research for melanoma prognosis as well [157].

The prognosis for melanoma has improved significantly with the rise of better diagnostic technologies, increased knowledge of molecular markers, and the advent of immunotherapies [158,159]. While the future of melanoma prognosis has displayed a rapid evolution with the incorporation of these advanced diagnostic tools and targeted therapies, these methods are not meant to replace conventional clinical and pathological assessments. Rather, they are intended to enhance the ability to create diagnostic and treatment plans that are more personalized for patients. GEP, in particular, has gained significant attention due to its ability to provide insight into the molecular makeup of a tumor, offering a more personalized risk assessment by predicting factors such as the likelihood of metastasis or recurrence. However, despite its growing popularity, the role of GEP in melanoma prognostication remains a topic of debate. Both the American Academy of Dermatology (AAD) and the NCCN have not yet fully endorsed GEP tests for routine clinical use, underscoring the need for further studies to validate their predictive reliability in melanoma outcomes [20,61]. As research continues, it is likely that these molecular techniques will become more integrated into the standard of care, paving the way for more personalized and precise melanoma treatments that can improve survival rates across all stages of melanoma.

## 5. Future Direction

Melanoma continues to pose a significant health challenge due to its increasing incidence and propensity for metastasis. The best way to improve outcomes is prevention through measures to reduce UV radiation exposure. The next best approach is early detection through better patient education and awareness and skin cancer screening. The development of non-invasive technologies such as ctDNA analysis and the emerging LC-OCT offer promising advancements in early detection and treatment monitoring. Although current GEP tests have shown promise in assessing prognosis, the greatest benefit would be to develop testing that predicts response to therapy and direct personalized care for patients with melanoma. The integration of other innovative imaging techniques and molecular diagnostics into clinical practice also holds great potential to enhance melanoma prognosis, reduce mortality rates, and improve personalized treatment strategies. While progress in melanoma diagnosis has been significant, concerns around the prognostic utility of some of the newer diagnostic tools, such as GEP tests, still remain. There is a need for more comprehensive research to validate the data linked to these new technologies and their role in improving long-term patient outcomes. Additionally, changing criteria for the classification of melanoma has led to altered treatment recommendations. For instance, following the American Joint Committee on Cancer 8th edition system, a wider variety of melanomas are now classified as stage IIIA, warranting adjuvant therapy. These new staging criteria have led to improved treatment outcomes [160].

The literature has shown that the treatment for melanoma has improved significantly since the advent of targeted therapies and immunotherapies nearly a decade ago, although challenges such as treatment resistance and disease recurrence continue to drive the need for novel approaches and combination regimens. Survival outcomes for melanoma patients with *BRAF* mutations have increased with management by BRAF and MEK inhibitors, and the use of combination therapies with these drugs has demonstrated even better overall survival rates with delayed resistance development. Immune checkpoint inhibitors targeting the PD-1/PD-L1 axis have also shown efficacy in treating both *BRAF*-mutant and wild-type melanomas. Targeted therapies and immunotherapies are both contributors to the increase in the number of long-term survivors, prompting questions regarding unmet needs in this growing demographic [161]. The recent application of neoadjuvant immunotherapy has shown much promise, and may soon replace the standard of care for the treatment of various melanoma pathologies [100]. While these treatments represent great advancements for treatment modalities for melanoma, challenges with resistance, toxicity, and limited penetration of some therapies remain and continue to underscore the need for ongoing research. With emerging therapeutic strategies like TILs, T-VEC, and TLR agonists, the challenge of overcoming resistance and enhancing immune responses against melanoma has recently been improved, and the combination of these novel approaches with radiotherapy and electrochemotherapy provides hope for improving patient outcomes.

Despite advancements in therapeutic options, improvements must be made regarding the speed at which therapies can be manufactured and delivered for patient use. This is particularly important for therapies that are personalized to each patient’s tumor microenvironment, such as melanoma vaccinations and TILs, which can take weeks to arrive [162]. In the meantime, efforts should be made to determine the best course of action during the manufacturing waiting period for these therapies. The trend in the literature also suggests that combination therapies are generally more effective than monotherapies for the treatment of melanomas. With the surge in new therapeutic options, significant emphasis should be placed on determining the most effective therapeutic combinations with the lowest rates of toxicity.

Artificial intelligence (AI) as an advancing technology is believed to be able to transform melanoma management in the near future (Figure 2). AI is being applied to classify histopathological images between nevi and melanoma, and mobile applications using AI have enhanced the ability to remotely to assess and monitor these lesions [163]. However, the reasoning behind an AI model’s “decision-making” has been met with suspicion among clinicians, who desire transparency for the set of conditions and criteria that determine the algorithms’ output. The lack of clear, interpretable decision-making methods of AI has significantly impeded the support of AI use in clinical practice [164]. Recently, a multimodal explainable AI (XAI) system has been developed to accurately diagnose melanoma by providing explanations for the AI’s output, as to close the interpretation gap felt by dermatologists [164]. AI is likewise being utilized in melanoma drug design, specifically to analyze extensive sets of biological data to identify potential therapeutic targets for melanoma [165,166]. As such, AI is accelerating the development of effective melanoma treatments by predicting drug efficacy and optimizing compound structures [165,166]. AI can contribute to the development of personalized treatment for patients with melanoma by analyzing data specific to the patient, such as genetic profiles and tumor characteristics. ML algorithms are then able to predict a patient’s response to various therapies, allowing for optimal treatment selection that maximizes efficacy and minimizes adverse effects. This personalized approach can enhance patient outcomes and represents a significant advancement in the treatment method and care for patients with melanoma [167,168]. As the landscape of melanoma treatment continues to expand, personalized therapies and combination approaches tailored to specific tumor profiles are likely to play a critical role in further decreasing resistance profiles and improving prognosis, thereby increasing overall survival rates and quality of life for patients with melanoma.

Further research on the integration of AI in the diagnosis, treatment, and prognosis of melanoma is needed to understand the full extent of its potential and limitations. While numerous studies have examined AI’s role in melanoma pathology, there remains a need for more comprehensive analyses of its clinical applications, effectiveness, and challenges [53]. Future research should focus on refining AI algorithms to improve accuracy, interpretability, and generalizability across diverse patient populations. Additionally, the ethical implications of AI-driven diagnostics, including bias in training data and the need for human oversight, should be critically evaluated. With a thorough understanding of these aspects, the field can move toward more reliable and clinically applicable AI-driven melanoma detection and management strategies.

In addition to these advancements, there is an emerging shift in mindset regarding the treatment of cancers which portrays tumors as an ecosystem of tumor subspecies competing for evolutionary dominance. This ecological disease perspective may have important implications in tumor management. For instance, aggressive therapy which aims to completely eliminate all tumor cells may put increased evolutionary pressure on the tumor, leading to resistance within months, as has been seen with BRAF/MEK inhibitors. An ecological approach may instead pursue therapies that aim to balance tumor cell subpopulations to prevent the emergence of therapy-resistant clones [169]. Additionally, just as introducing beneficial species into an ecosystem can restore balance, there is evidence to suggest that introducing bacteria into the gut microbiome via fecal microbiota transplants can help to enhance antitumor immunity in melanoma patients [170]. Furthermore, tumor budding can be likened to ecological islands, geographically separated from the mainland and under unique evolutionary pressures as they interact with stromal cells and immune infiltrates at the tumor–host interface. Given that budding enhances a tumor’s ability to metastasize, therapies which alter the terrain in and around buds, perhaps by targeting the extracellular matrix, may expose melanoma to increased immune attacks and prevent metastasis, just as modifications to an ecosystem’s terrain may affect migration patterns [169]. Further consideration of melanoma as an ecological disease may lead to promising advancements in tumor management going forward.

## 6. Conclusions

Recent developments made in the diagnosis, treatment, and prognosis of melanoma have led to improved patient survival rates and length of survival post treatment. With advancements in targeted therapies, the combination of BRAF and MEK inhibitors has demonstrated prolonged survival and delayed resistance in patients with *BRAF* mutations, while immune checkpoint inhibitors have shown effectiveness across both *BRAF*-mutant and wild-type melanoma cases. Emerging therapies, including TILs, TLRs, and gene therapies, are providing hope for addressing resistance mechanisms and enhancing immune-mediated tumor control. Additionally, the growing interest in neoadjuvant therapy highlights its potential in strengthening antitumor immune responses pre-surgery, which may reduce surgical morbidity and improve outcomes.

Melanoma research has made significant strides, but challenges persist. While personalized diagnostic tools show promise in predicting melanoma progression, more research is required to confirm their long-term benefits. Additionally, therapeutic advancements have improved patient outcomes, but treatment resistance and toxicity continue to drive the need for more refined therapies. Combination therapies and personalized treatment approaches offer an avenue by which to further reduce melanoma-related mortality and enhance individualized patient care. Future efforts should focus on optimizing treatment timing, minimizing toxicity, and developing rapid delivery mechanisms for therapies.

## Figures and Tables

**Figure 1 cancers-17-00707-f001:**
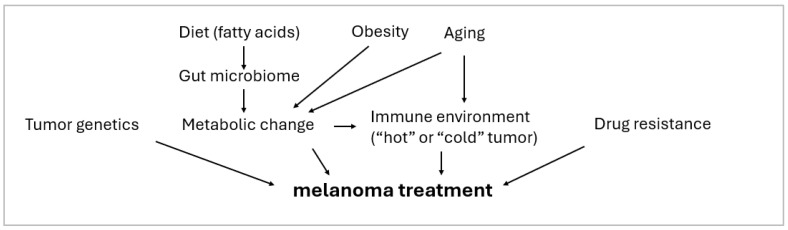
Factors that affect melanoma treatment.

**Figure 2 cancers-17-00707-f002:**
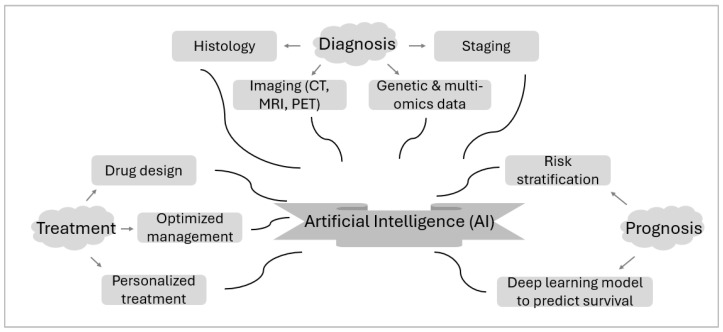
Artificial intelligence in melanoma management.

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
