# Peer review of "Clinical Approaches for the Management of Skin Cancer: A Review of Current Progress in Diagnosis, Treatment, and Prognosis for Patients with Melanoma"

_cancers, 2025, doi:10.3390/cancers17040707_

Round 1
Reviewer 1 Report (Previous Reviewer 2)
Comments and Suggestions for Authors
The authors have addressed all of the above recommendations to improve the manuscript. It is now a well-written review that provides a global overview of the state of the art in the diagnosis and treatment of malignant melanoma.
I recommend acceptance.
Author Response
Thanks for your review. We have compiled all the responses in one file and attached.

Reviewer 2 Report (Previous Reviewer 3)
Comments and Suggestions for Authors
The topic of diagnosis and treatment of melanoma is continuosly evolving; this review contains the most relevant news of the latest years. I think it could be enough.
Author Response
Thanks for your review. We have compiled all the responses in one file and attached.

Reviewer 3 Report (New Reviewer)
Comments and Suggestions for Authors
Melanoma continues to be a leading cause of skin-cancer-related mortalities worldwide, with incidence rates on the rise. This review elucidates the advancements in the multidisciplinary management of melanoma, emphasizing the critical need for ongoing research to enhance patient outcomes.Several key points should be considered to further enhance the quality of this paper as below,
1) Since the title “Clinical Approaches for the Management of Skin Cancer...” , at the beginning of the Introduction section, the types of skin cancer should be clearly delineated. For instance, skin cancer can be categorized into cutaneous melanoma and non-melanoma skin cancer (NMSC).
2) Line 80-81,“This review will address the recent progress made in the areas of diagnosis, treatment, and prognosis for patients with melanoma.”, what methods were used when the comprehensive literature search was carried out, e.g. PubMed, Scopus, Web of Science, and Google Scholar databases? The search terms used for the literature review either include or exclude specific criteria?
3) Line 106 “2.1.1 Digital Dermoscopy”, what about AI used in dermoscopic image analysis for melanoma (https://pubmed.ncbi.nlm.nih.gov/39806282/).
4) Line 192, “ctDNA is originated from circulating cancer cells (CTC)”, are you sure that ctDNA is not from other sources, only CTC?
5) In line 405,“3.8. Radiotherapy”, a recent paper proposes a potential "Anti-Warburg Effect" (AWE) in circulating tumor cells (CTCs), representing a metabolic shift that bridges primary tumors and metastases. This observed AWE may hold significant clinical importance, as it is strongly correlated with therapeutic response in melanoma patients (https://pubmed.ncbi.nlm.nih.gov/38300633/). This finding warrants further discussion.
6) Lie 491, “5. Future Direction”, more studies on the application of artificial intelligence (AI) in melanoma research should be incorporated and thoroughly discussed. Given the extensive body of literature on this topic, such as AI's role in melanoma pathology, it is crucial to explore these advancements in greater depth.
7) Emerging data suggest that cancer can be conceptualized as an ecological disease (https://pubmed.ncbi.nlm.nih.gov/37056571/). It would be valuable to review this perspective and explore how the ecological approach can be integrated into melanoma research in the "Future Directions" section.
Author Response
Thanks for your review. We have compiled all the responses in one file and attached.

Round 2
Reviewer 3 Report (New Reviewer)
Comments and Suggestions for Authors
The authors have carried out excellent revisions in response to these issues, and this article has well met the requirements for publication.
This manuscript is a resubmission of an earlier submission. The following is a list of the peer review reports and author responses from that submission.
Round 1
Reviewer 1 Report
Comments and Suggestions for Authors
This is a manuscript that reviews the latest advances in diagnosis, treatment and prognosis in melanoma.
I understand that it is not intended to be a guide, so the indications for the different specific therapies are not discussed. For example, the indication of neoadjuvant or adjuvant immunotherapy is not discussed.
Its statements are correct and so is the literature on which they are based.
Understood as a review article, it is suitable for publication.
Not being an English-native speaker, I am not qualified to review the language.
Author Response
We appreciate the reviewers' work. Responses to all the reviewers' comments have been complied in one file and attached.

Reviewer 2 Report
Comments and Suggestions for Authors
The authors provided a comprehensive review of clinical approaches to advances in the diagnosis, treatment and prognosis of melanoma.
However, I miss some balanced remarks regarding newer diagnostic tools that histology and immunohistology represent the gold standard.
Also missing is the brilliant work by Bastian et al in dissecting the genetic landscape of melanoma.
I also miss a statement on where we are after more than 10 years of targeted therapy and checkpoint therapy.
The hottest topic in melanoma therapy, the neoadjuvant approach, is not discussed. Blank et al ASCO 2024.
I have several citations that should be included in the manuscript.
Schadendorf D, Dummer R, Flaherty KT, Robert C, Arance A, de Groot JWB, Garbe C, Gogas HJ, Gutzmer R, Krajsová I, Liszkay G, Loquai C, Mandalà M, Yamazaki N, Queirolo P, Guenzel C, Polli A, Thakur M, di Pietro A, Ascierto PA. COLUMBUS 7-year update: a randomised, open-label, phase III trial of encorafenib plus binimetinib versus vemurafenib or encorafenib in patients with BRAF V600E/K-mutated melanoma. Eur J Cancer. 2024 Jun;204:114073. doi: 10.1016/j.ejca.2024.114073. Epub 2024 Apr 24. PMID: 38723373.
Schroeder C, Gatidis S, Kelemen O, Schütz L, Bonzheim I, Muyas F, Martus P, Admard J, Armeanu-Ebinger S, Gückel B, Küstner T, Garbe C, Flatz L, Pfannenberg C, Ossowski S, Forschner A. Tumour-informed liquid biopsies to monitor advanced melanoma patients under immune checkpoint inhibition. Nat Commun. 2024 Oct 9;15(1):8750. doi: 10.1038/s41467-024-52923-0. PMID: 39384805; PMCID: PMC11464631.
Amaral T, Nanz L, Stadler R, Berking C, Ulmer A, Forschner A, Meiwes A, Wolfsperger F, Meraz-Torres F, Chatziioannou E, Martus P, Flatz L, Garbe C, Leiter U. Isolated melanoma cells in the sentinel lymph node in stage IIIA melanoma correlate with a favourable prognosis similar to stage IB. Eur J Cancer. 2024 Apr;201:1139-12. doi: 10.1016/j.ejca.2024.1139-12. Epub 2024 Feb 10. PMID: 38368742.
Reitmajer M, Leiter U, Nanz L, Amaral T, Flatz L, Garbe C, Forschner A. Long-term survival of patients with stage IV melanoma: evaluation of 640 patients with stage IV melanoma between 2014 and 2017. J Cancer Res Clin Oncol. 2024 Jan 18;150(1):15. doi: 10.1007/s00432-023-05533-0. PMID: 38238578; PMCID: PMC10796594.
Wolchok JD, Chiarion-Sileni V, Rutkowski P, Cowey CL, Schadendorf D, Wagstaff J, Queirolo P, Dummer R, Butler MO, Hill AG, Postow MA, Gaudy-Marqueste C, Medina T, Lao CD, Walker J, Márquez-Rodas I, Haanen JBAG, Guidoboni M, Maio M, Schöffski P, Carlino MS, Sandhu S, Lebbé C, Ascierto PA, Long GV, Ritchings C, Nassar A, Askelson M, Benito MP, Wang W, Hodi FS, Larkin J; CheckMate 067 Investigators. Final 10-year results with nivolumab plus ipilimumab in advanced melanoma. N Engl J Med. 2024 Sep 15. doi: 10.1056/NEJMoa2407417. Epub ahead of print. PMID: 39282897.
Long GV, Hauschild A, Santinami M, Kirkwood JM, Atkinson V, Mandala M, Merelli B, Sileni VC, Nyakas M, Haydon A, Dutriaux C, Robert C, Mortier L, Schachter J, Schadendorf D, Lesimple T, Plummer R, Larkin J, Tan M, Adnaik SB, Burgess P, Jandoo T, Dummer R. Final results of adjuvant dabrafenib plus trametinib in stage III melanoma. N Engl J Med. 2024 Nov 7;391(18):1709-1720. doi: 10.1056/NEJMoa2404139. Epub 2024 Jun 19. PMID: 38899716.
Ascierto PA, Casula M, Bulgarelli J, Pisano M, Piccinini C, Piccin L, Cossu A, Mandalà M, Ferrucci PF, Guidoboni M, Rutkowski P, Ferraresi V, Arance A, Guida M, Maiello E, Gogas H, Richtig E, Fierro MT, Lebbe C, Helgadottir H, Queirolo P, Spagnolo F, Tucci M, Del Vecchio M, Cao MG, Minisini AM, De Placido S, Sanmamed MF, Mallardo D, Paone M, Vitale MG, Melero I, Grimaldi AM, Giannarelli D, Dummer R, Sileni VC, Palmieri G. Sequential immunotherapy and targeted therapy for metastatic BRAF V600-mutant melanoma: 4-year survival and biomarker evaluation from the phase II SECOMBIT trial. Nat Commun. 2024 Jan 2;15(1):146. doi: 10.1038/s41467-023-44475-6. PMID: 38167503; PMCID: PMC10761671.
What is the final conclusion of all the recent achievements?
Author Response

(The authors gave the same response as above.)

Reviewer 3 Report
Comments and Suggestions for Authors
Dear Authors, section about dermoscopy should be improved with recent findings in early diagnosis of melanomas with a longitudinal diameter < 6 mm (Nazzaro G, et al. Int J Dermatol 2023).
About diagnostic tools in melanomain situ and lentigo maligna, there are recent reviews from which you could take information (Karponis D, el al. Cutaneous melanoma in situ: a review. Clin Exp Dermatol. 2024; Gupta S, et al. Lentigo Maligna Part I: Epidemiology, Risk Factors, and Diagnosis. J Am Acad Dermatol. 2024 )
Please correct "mortality" in page 2.
The paragraph about OCT is less relevant in my opinion as this tool is not routinely used for melanoma diagnosis, while reflectance confocal microscopy has more evidence but no widely diffused. You could add some suggestions about the role of reflectance confocal microscopy in facial lesions.
3.2 please correct "immunotherapy"
In this paragraph you should mention the recent approval of pembrolizumab in STAGE II melanoma and the studies about nivolumab in stage II melanoma
Author Response

(The authors gave the same response as above.)
